# Enantiospecific electrochemical rearrangement for the synthesis of hindered triazolopyridinone derivatives

Zenghui Ye [1,2], Yanqi Wu[3], Na Chen[1], Hong Zhang[1], Kai Zhu[1,2], Mingruo Ding[1], Min Liu[1], Yong Li[1] & Fengzhi Zhang [1,2✉]

Triazolopyridinone derivatives are of high value in both medicinal and material chemistry. However, the chiral or hindered triazolopyridinone derivatives remain an underexplored area of chemical space because they are difficult to prepare via conventional methods. Here we report an electrochemical rearrangement for the efficient synthesis of otherwise inaccessible triazolopyridinones with diverse alkyl carboxylic acids as starting materials. This enables the efficient preparation of more than 60 functionalized triazolopyridinones under mild conditions in a sustainable manner. This method is evaluated for the late stage modification of bioactive natural products, amino acids and pharmaceuticals, and it is further applied to the decagram scale preparation of enantiopure triazolopyridinones. The control experiments support a mechanism involving an oxidative cyclization and 1,2-carbon migration. This facile and scalable rearrangement demonstrates the power of electrochemical synthesis to access otherwise-inaccessible triazolopyridinones and may find wide application in organic, material and medicinal chemistry.

[1] College of Pharmaceutical Science, Zhejiang University of Technology, 310014 Hangzhou, PR China. [2] Collaborative Innovation Center of Yangtze River Delta Region Green Pharmaceuticals, Zhejiang University of Technology, 310014 Hangzhou, PR China. [3] Zhejiang University of Technology, 310014 Hangzhou, PR China. ✉email: zhangfengzhi@zjut.edu.cn

Triazolopyridinone derivatives are of high value for different applications in materials[1], pharmaceuticals[2–4], or agrochemicals[4] (Fig. 1a). The primary substituted triazolopyridinone derivatives are widespread and can be prepared by conventional $S_N2$ substitution from the corresponding triazolopyridinones and halides (Fig. 1b). However, there are only scarce reports about the synthesis of secondary substituted triazolopyridinone derivatives due to the synthetic challenge associated with elimination side reactions or steric issues, and there are only two articles reported for the access of enantiomerically enriched triazolopyridinones using Mitsunobu (only three examples, no enantiomeric excess (ee) values reported)[5] or aza-Michael (only one example in 77% yield and 89% ee, a 2-day reaction)[6] conditions, respectively. Moreover, there is no report about the synthesis of sterically hindered tertiary substituted triazolopyridinone derivatives, which still remain underexplored area of chemical space.

Alkyl carboxylic acids are ubiquitous in every aspect of chemistry, and are widely used as building blocks for chemical synthesis because of their stability, convenience for use, and commercial availability in low cost and diverse structures[7–12]. Previously, alkyl carboxylic acids have been well explored for the Kolbe-type electrolysis[13] and decarboxylation via activated Barton or N-hydroxyphthalimide esters[14–22], which could generate alkyl radicals or carbocations for further coupling reactions (Fig. 1c). However, the atom economy of Barton type decarboxylation is low because of the requirement of activating groups, which were discarded as waste after the reaction. Very recently, Baran et al. reported an elegant synthesis of hindered dialkyl ethers with electrogenerated carbocations generated from alkyl carboxylic acids[23], which demonstrates the power of electrochemistry to substantially improve the synthetic efficiency[24–34]. Rearrangement reactions through a molecular skeletal reorganization are powerful chemical transformations for generating

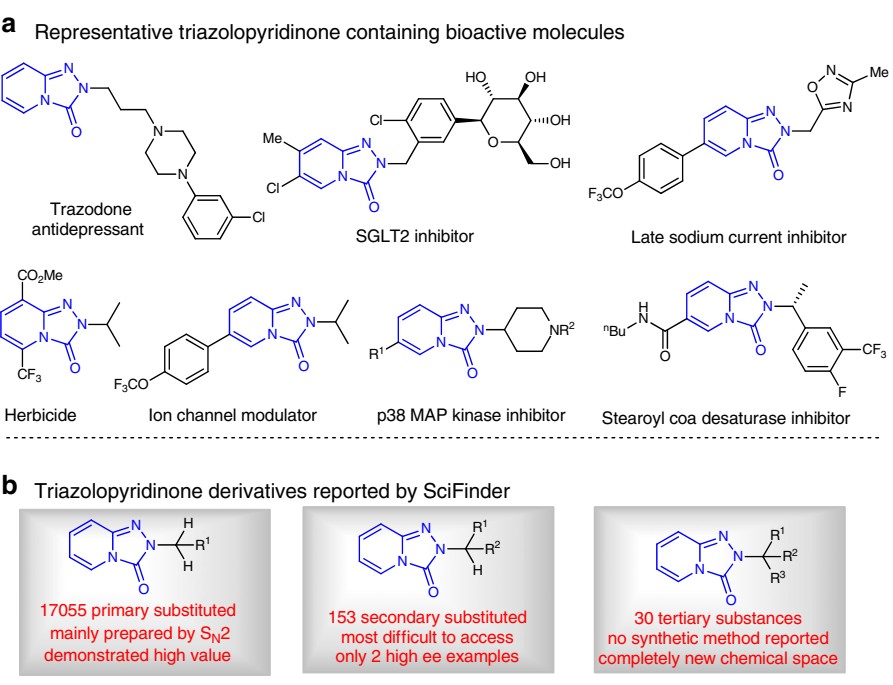

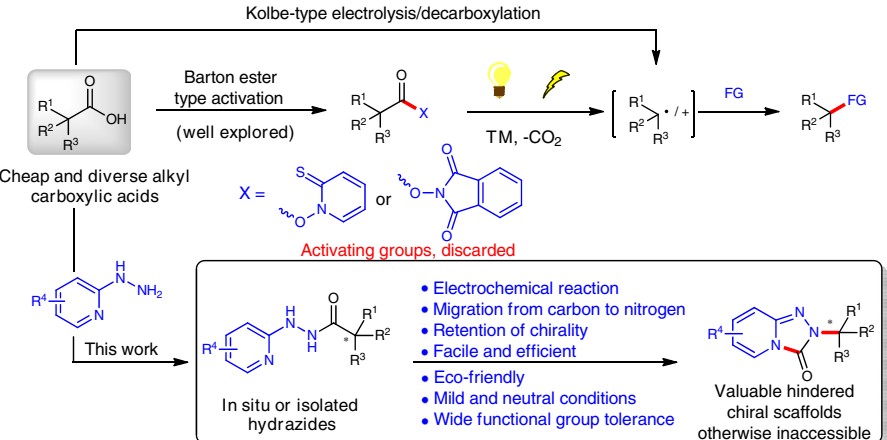

**Fig. 1 Background and reaction development. a** Representative bioactive molecules with triazolopyridinone scaffold. **b** Triazolopyridinone derivatives reported by SciFinder. **c** Alkyl carboxylic acid activation modes. TM transition metal, FG functional group.

structural complexity and diversity[35–39]. Based on our research experience in the development of C–C or C–X bond forming methodologies via electrochemistry[40–42], we envisioned that the in situ or isolated hydrazides, prepared from diverse alkyl carboxylic acids and 2-hydrazinopyridines, might undergo an atom/step-economical rearrangement to afford those otherwise-inaccessible triazolopyridinone derivatives under metal and external oxidant free electrochemical conditions (Fig. 1c). Here, we show that an electrochemical rearrangement involving a tandem oxidative cyclization and 1,2-carbon migration was developed for the efficient synthesis of valuable (enantiopure) triazolopyridinone derivatives.

## Results

**Reaction optimization.** As a proof-of-principle study, we began our condition optimization with the isolated hydrazide as substrate, which was readily prepared from 2-phenylacetic acid and 2-hydrazinopyridine (see Supplementary Table 1). After extensive screening, it was found the reaction in a simple undivided cell led to the desired product **1** in 99% yield (graphite anode, platinum cathode, tris(*p*-bromophenyl)amine ((4-Br-C₆H₄)₃N, 10 mol%) as the redox catalyst, tetrabutylammonium tetrafluoroborate as the supporting electrolyte, MeOH, 70 °C), and the structure of triazolopyridinone product **1** was further confirmed by single crystal X-ray diffraction analysis (Fig. 2).

**Evaluation of substrate scope.** With optimized conditions in hand, the substrate scopes of this rearrangement reaction were evaluated as shown in Fig. 2. First, inexpensive primary carboxylic acids with a benzyl substituent attached to the acid functionality were tested, and all gave the corresponding products in moderate to excellent yields (**2–22**). The substrates with electron-donating groups on the benzene ring (**2–4, 12–15**) afforded the corresponding products in much higher yields than those with electron-withdrawing groups (**5–8, 10–11**). The substrates with a naphthyl, thienyl, or allylic substituent were also effective, albeit in lower yields (**22–24**). Next, the secondary alkyl carboxylic acids were tested (**25–34**). The substrates with either alkyl or phenyl substituents at the benzyl position all gave the corresponding products in excellent yields (**25–29**). Substrates with a heteroatom at the α-C were also effective although the yields dropped, probably because of the inductive effect of the heteroatom (**30–34**). The tertiary carboxylic acids such as simple alkyl substituted (**35–37**), deuterated (**38**), fluorinated (**39**), and cyclic systems (**41–44**) were demonstrated effective as well. It is worth to mention that these tertiary substituted triazolopyridinone derivatives had never been reported before, and they cannot be prepared by the conventional nucleophilic substitution from the corresponding 1,2,4-triazolo[4,3-*a*]pyridin-3(2*H*)-one and alkyl halides (see Supplementary methods). Finally, this protocol was successfully applied for the late stage modification of pharmaceuticals (**45–50**), amino acids (**51–52**), and natural products (**53–54**), which would provide an efficient and fast way to prepare their analogs for structure and activity relationship studies in medicinal chemistry. As demonstrated above, this reaction exhibits broad functional group tolerance such as aryl and alkyl halides (F: **5, 39** and **46**; Cl: **6, 16, 20, 27, 37, 45,** and **49**; Br: **7**; I: **17**), ethers (**3–4, 12–13, 30–33,** and **49–50**), esters (**19, 30,** and **54**), ketone (**47–48**), enone (**54**), olefin (**24**), protected amines (**34, 43, 45, 51–52**), etc., which provide functional handles for further diverse transformations. This protocol also can be conducted without isolation of the hydrazide though the overall yields (in the brackets) are a little bit lower. It worth to mention that these drug-like triazolopyridinone derivatives have been reported with a wide range of biological activities

such as antidepressant, antidiabetic, glycogen synthase kinase-3 inhibitor, late sodium current inhibitor, herbicidal, etc[2,3,43–48]. The successful development of this rearrangement would inspire more exploration on their chemical space and applications.

**Synthetic applications.** A series of optically pure hydrazide substrates were then synthesized from chiral acid containing molecules including amino acids (**51** and **52**) and best-selling drugs (**55–60**) to verify whether the configuration could be retained in this rearrangement if the migrating terminus has a stereocenter (Fig. 3a). As a result, the corresponding cyclized products were obtained under the standard conditions with high yields and excellent *ee*. The stereochemistry of (*S*)-**59** was determined unambiguously by X-ray crystallographic analysis, which indicate the products were obtained with retention of configuration. It is worth to mention that this is one of the few rearrangements, which can retain the stereochemistry successfully, and those drug-like (chiral) enantiopure triazolopyridinone derivatives are difficult to make by conventional methods, and are of high value for drug discovery.

To further demonstrate the synthetic utility of this protocol, we first applied this rearrangement as key step for the synthesis of stearoyl-CoA desaturase inhibitor analog **63** (Fig. 3b)[47]. Under the standard conditions the hydrazide **61** was converted to the key intermediate **62** in 70% yield, which was further transformed into **63** by hydrolysis and condensation in 97% overall yield. Next, we applied it for the synthesis of hindered triazolopyridinone compound **67**, which is an otherwise-inaccessible analog of top-selling drug antidepressant trazodone (Fig. 3c). The chloride **65** was prepared efficiently from hydrazide **64** under the standard electrochemical conditions, which was then coupled with the piperazine compound **66** to give the trazodone analog **67** successfully. It was indicated that this methodology could be used for the efficient synthesis of previously inaccessible trazodone analogs[3]. Finally, the gram or even decagram scale reaction of **68** was conducted and the desired enantiopure product (*S*)-**60** was readily obtained in good yield and excellent *ee* (Fig. 3d), which demonstrated the scalability of this powerful rearrangement.

**Mechanistic studies.** In order to better understand the mechanism of this rearrangement, a series of control experiments were conducted (Fig. 4). First, a crossover experiment with a mixture of **69** and **70** as the substrates was conducted, and it was found that no crossover products were obtained. This experiment demonstrated the rearrangement must be an intramolecular reaction (Fig. 4a). Second, various additives were tested for this rearrangement with hydrazide **71** as substrate, and it was found the addition of radical scavengers (BHT and TEMPO), H-atom donors (triethylsilane or 1,4-cyclohexadiene) or acetic acid did not inhibit the reaction (Fig. 4b). Third, the radical-clock experiments with **72** and **75** as the substrates gave the corresponding rearranged products **73** and **76**, respectively without the detection of ring forming or opening products **74** and **77**, which indicated this rearrangement might not involve a radical mechanism (Fig. 4c). Then, with hydrazide **78** as substrate the cyclized product **79** was obtained without detection of 1,2-hydride shift product **80**, which showed the stepwise carbocation formation might not operate for this rearrangement (Fig. 4d). Finally, with hydrazide **81** as substrate, it was found that a diazo byproduct **83** was isolated in 27% yield along with the rearrangement product **82** in 58% yield with the hydrolysis of phthalimide ring (Fig. 4e). Interestingly, treatment of the diazo compound **83** in MeOH under reflux for 24 h, gave the

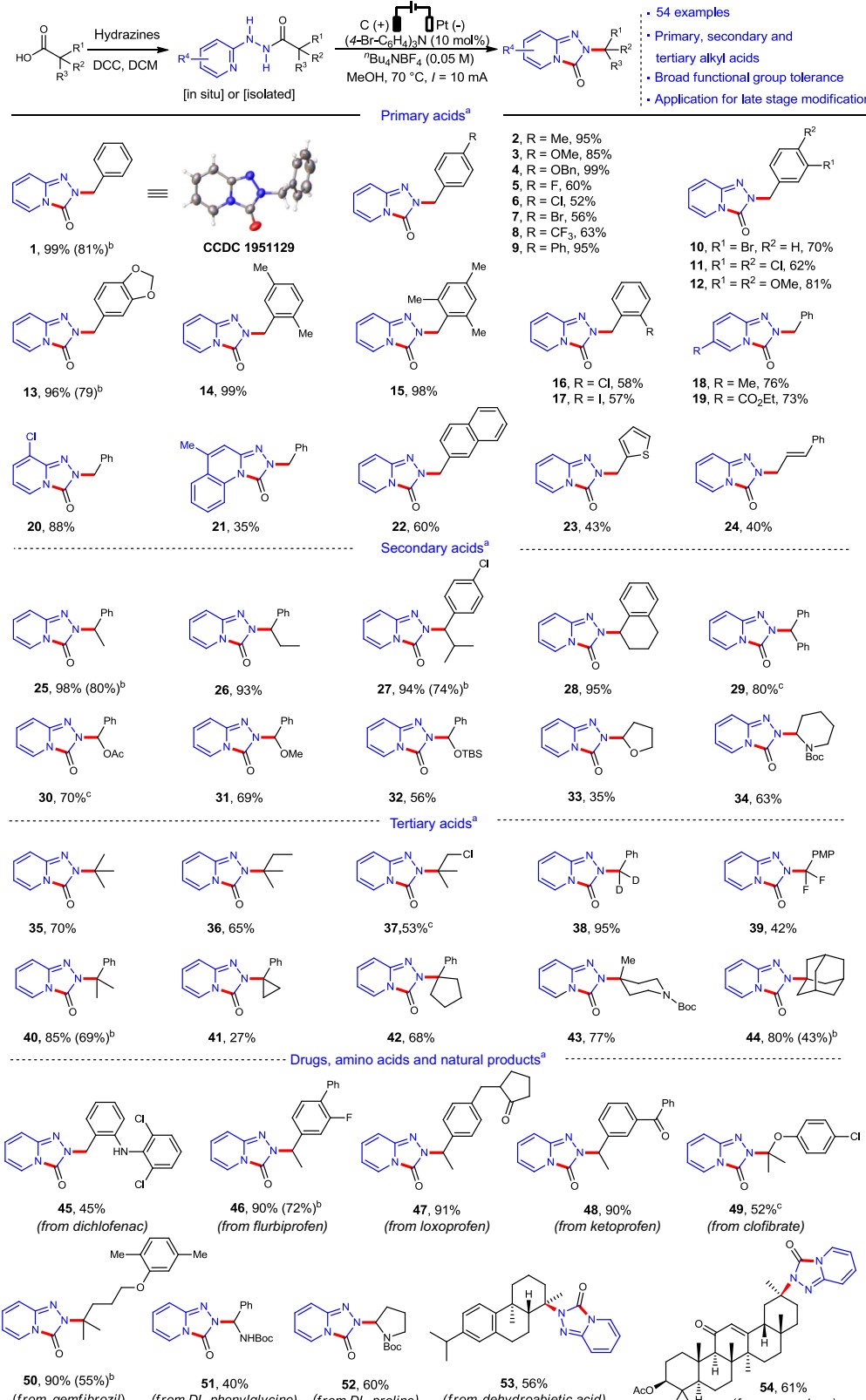

**Fig. 2 Substrate scope and late stage modifications via electrochemical rearrangement.** [a]Reaction conditions: undivided cell, graphite rod ($\phi$ 6 mm), Pt cathode (1 cm × 1 cm), hydrazide (0.3 mmol), $^{n}$Bu$_4$NBF$_4$ (0.5 mmol), MeOH (10 mL), 70 °C, 1.8 h (2.2 F/mol); [b]Yield for one-pot reaction with in situ hydrazide; [c]40 °C. Ac acetyl, Boc *tert*-butyloxycarbonyl, PMP 4-methoxyphenyl, TBS *tert*-butyldimethylsilyl.

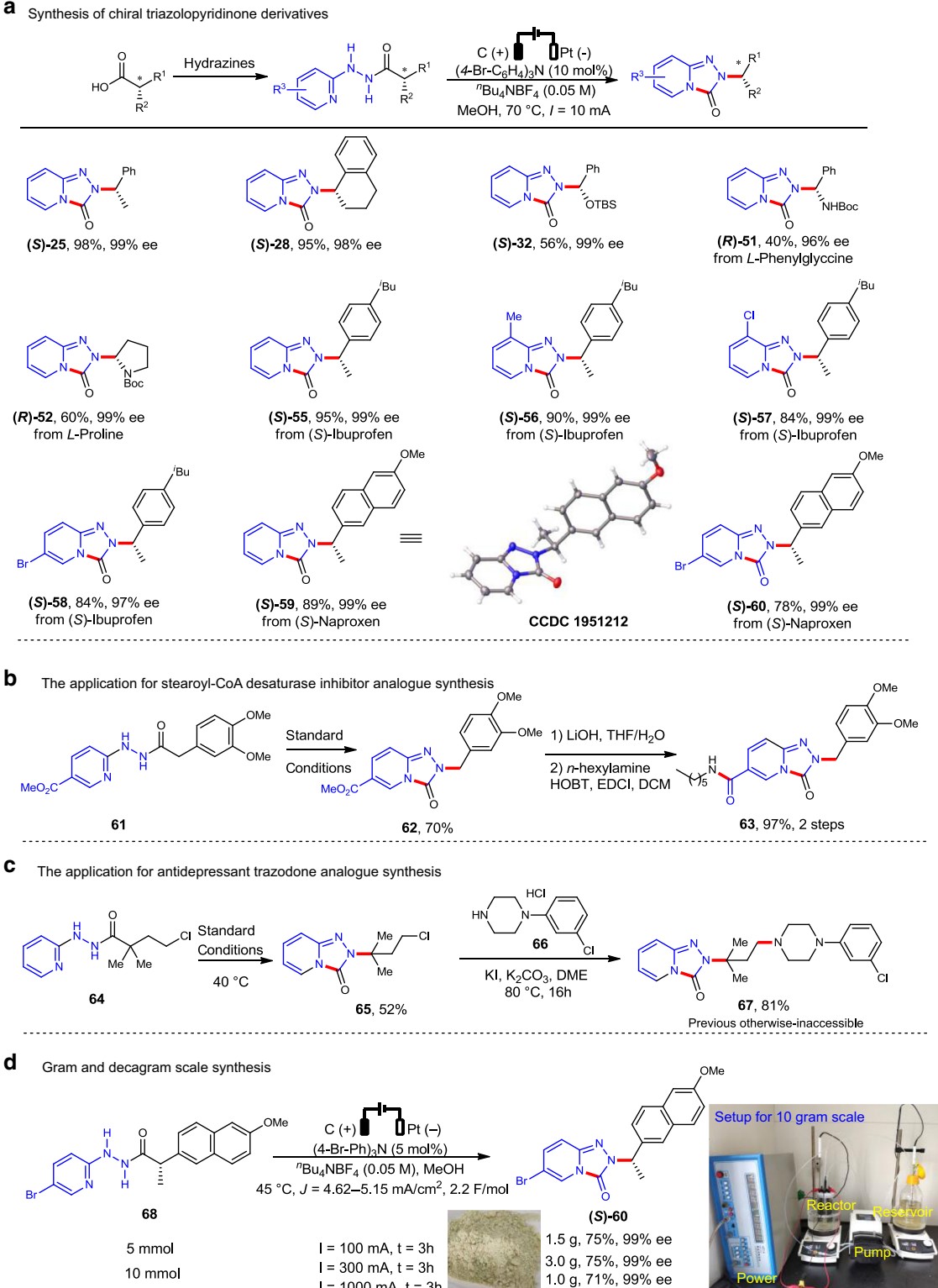

**Fig. 3 Synthetic applications of this electrochemical rearrangement. a** Synthesis of chiral triazolopyridinone derivatives. **b** Synthesis of stearoyl-CoA desaturase inhibitor analog. **c** Synthesis of antidepressant drug trazodone analog. **d** Large scale synthesis.

rearrangement product **82** in 79% yield, which indicated that this rearrangement might take place via the diazo intermediate. Based on the above control experiments and cyclic voltammetry experiments, a reaction mechanism was proposed: by electrochemical oxidation, a trans-diazo intermediate might form which

would readily interconvert with its cis-diazo species because of low energy barrier. Followed by the nucleophilic addition of nitrogen to the carbonyl group and a 1,2-alkyl shift from carbon to nitrogen would afford the desired cyclized product (see Supplementary Fig. 18).

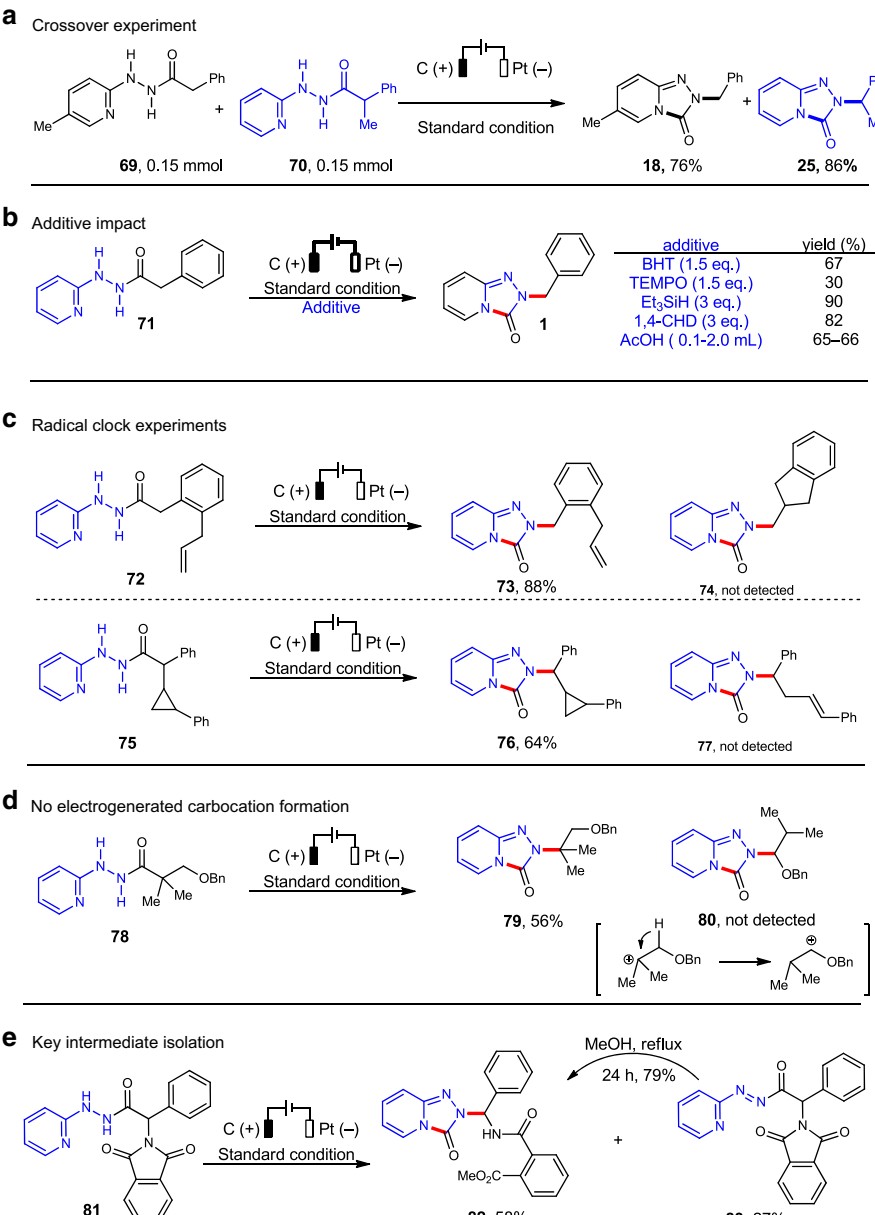

**Fig. 4 Control experiments and mechanistic probes. a** Crossover experiments. **b** Impact of additives. **c** Radical-clock experiments. **d** Control experiment to demonstrate no electrogenerated carbocation formation. **e** Key intermediate isolation.

This activation mode of carboxylic acids opens the door to convert the diverse and cheap alkyl carboxylic acids into otherwise-inaccessible high value triazolopyridinone derivatives via a highly sustainable electrochemical rearrangement. It is anticipated that this powerful rearrangement will inspire more new findings and find various applications in future.

## Methods

**General procedure for the electrochemical rearrangement reaction**. Further experimental details are provided in the Supplementary Information.

General procedure for the electrochemical synthesis of compound **1**. A 10-mL three-necked round-bottomed flask was equipped with a graphite carbon anode ($\phi$ 6 mm, about 1 cm immersion depth in solution), a platinum plate (1 cm × 1 cm) cathode and a stirring bar. The flask was charged with hydrazide **71** (0.3 mmol, 1 equiv.), tris(*p*-bromophenyl)amine (0.03 mmol, 10 mol%), $^{n}Bu_4NBF_4$ (0.5 mmol), and MeOH (10 mL). The reaction mixture was stirred and electrolyzed at a constant current of 10 mA under 70 °C for 1.8 h (2.2 F/mol). When the reaction was finished, the reaction mixture was transferred to a single-necked flask and concentrated under reduced pressure. The resulting residue was washed with water and extracted with EtOAc (3 × 20 mL). The combined organic solution was dried over anhydrous $Na_2SO_4$ and concentrated under reduced pressure. The given crude was purified by column chromatography through silica gel to provide the desired product **1** in 99% yield. Full experimental details and characterization of new compounds can be found in the Supplementary Information.

## Data availability

The X-ray crystallographic coordinates for structures reported in this article have been deposited at the Cambridge Crystallographic Data Centre (CCDC), under deposition numbers CCDC 1951129 (**1**) and 1951212 (**S-59**). These data can be obtained free of charge from The Cambridge Crystallographic Data Centre via http://www.ccdc.cam.ac.uk/data_request/cif. The data supporting the findings of this study are available within the article and its Supplementary information files. Any further relevant data are available from the authors on request.

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

## Acknowledgements

This research was funded by National Natural Science Foundation of China (NSFC, 21871234), and Zhejiang Provincial NSFC for Distinguished Young Scholars (LR15H300001). We thank Profs. Yongqiang Tu (Lanzhou Univ.), Jieping Zhu (EPFL), Matthew Gaunt (Cambridge Univ.), and John Moses (La Trobe Univ.) for helpful discussion.

## Author contributions

Z.Y., Y.W., and F.Z. conceived the project, designed the experiments, and analyzed the data. Z.Y., N.C., H.Z., K.Z., M.D., M.L., and Y.L., performed the experiments. F.Z. wrote the paper with the help of co-authors.

## Competing interests

The authors declare no competing interests.
