## [Peer Review File · Nature Communications]

Reviewers' comments:

Reviewer #1 (Remarks to the Author):

The submitted manuscript reports an electrochemical synthesis of N-alkyl triazolopyridinones from hydrazides of 2-hydrazinopyridine using a triarylamine as an electrochemical mediator, with around 60 reported examples. The triazolopyridine motif is found in a number of synthetic bioactive molecules that are of interest in drug discovery. To date, the synthesis of N-alkyl triazolopyridine containing compounds has been limited mostly to sterically unencumbered nitrogen substituents, installed largely through N-alkylation using primary alkyl electrophiles. The electrochemical synthesis described in the current work is effective for a variety of primary, secondary and tertiary N-substituents. Significantly, enantiomerically enriched hydrazide precursors possessing stereogenic centers alpha to the carbonyl group undergo stereospecific migration of the chiral group with retention of stereochemistry. The reaction mechanism is intriguing, and the authors propose that the hydrazide intermediate is oxidized to the diazo intermediate (shown as trans stereoisomer), which they propose, cyclizes to a pyridinium intermediate. This undergoes a 1,2-shift (C to N) of the acyl (COR) R substituent. A reactions were carried out to exclude mechanistic pathways involving radical, or carbocation intermediates. In one case an acyl diazo intermediate was isolated as a reaction side product, and shown to undergo rearrangement to the corresponding triazolopyridinone. In the electrochemical step, the two-electron oxidation is proposed to be mediated by the triarylamine. The process was performed on scales up to ~10 g, which should be attractive to those wishing to apply the methodology.

Overall, this is an intriguing electrochemical transformation, which should be of broad interest to organic chemists, particularly synthetic chemists and medicinal chemists. There are very significant elements of novelty and originality in the work, and the results would not have been easy to predict (I was not able to find related transformations). The mechanistic discussion is somewhat limited in the main paper, and the proposed cyclization of a trans-diazo species deserves further comment (barrier to cis/trans isomerization, or addition-elimination of MeOH?). While I was not entirely convinced by the mechanistic proposal, I am not able to propose a more convincing one. The role of the mediator is not commented on in the main paper, and it is not clear if the mediator or the supporting electrolyte are recovered. These are significant points, and as the manuscript stands, the claims of eco friendly and sustainable should certainly be removed. However, that does not detract from the interest and novelty of the work, which I am sure will influence others working in electrochemistry.

Overall, the paper should be published in Nature Communications subject to addressing some comments noted below.

I did not see any comment from the authors on N-aryl triazolopyridinones. Are these accessible using the reported method?

Abstract, line 18: "This enables the atom- and step-economical preparation of more than 60 functionalized triazolopyridinone derivatives under mild conditions in a sustainable manner."

This statement is not really correct. The preparation of the starting materials is not atom-economical, so the whole sequence can not be claimed to be atom-economical, even if the electrochemical synthesis is. The particular example of electrochemical synthesis is actually not that great in terms of sustainability either, due to the large excess of added electrolyte included in the electrolysis. Furthermore, is the mediator recovered? It is a high molecular weight compound, so the mass used is probably not insignificant, even at 10 mol %.

Line 44-46, page 2. Improve accuracy of statement: "However, the atom-economy of these cross-coupling reactions is low because of the requirement of activating groups which were discarded as waste after the reaction." The previous sentence describes Kolbe and Barton procedures for radical generation. The statement is certainly true for the Barton method, but less accurate for Kolbe (although it does lose CO₂). The next sentence, referring to Barton's work on "electro generated carbocations from carboxylic acids" (ref 40), is then described as sustainable (this is also decarboxylative, but also contains additives such as silver salts).

In Fig 1. B. Can the authors clarify whether they are referring to "2 chiral" or enantiomerically enriched examples? Please consider the use of the term "chiral".

Figure 1: Is the process electrocatalysis? It is an electrochemical reaction mediated by an electrocatalyst.

Figure 1: Is the process eco friendly? The electrochemical process uses 5:3 ratio of supporting electrolyte to substrate, which is quite high. Was the Bu₄NBF₄ recovered after the reaction? The electrochemical reaction is also quite dilute (0.03 M), which requires quite a lot of solvent when scaled up. The triarylamine mediator is used in 10 mol %, but is that recovered?

Line 88, page 4: "This protocol also can be conducted in a simple one-pot reaction without isolation of the hydrazide though the overall yields (in the brackets) are a little bit lower." The procedure described is not "one pot"; the authors simply took the crude material (containing the urea byproduct from DCC) from the first coupling reaction into the electrochemical step (solvent change and reaction vessel change).

Line 119, page 6: "A library of previously inaccessible trazodone analogues could be made efficiently by this strategy, and further evaluated for the binding affinity for some subtypes of serotonin receptors in search of novel drugs for the treatment of depression and anxiety." Suggest that this is revised to indicate that the methodology could be used for the synthesis of previously inaccessible trazodone analogues.

Although the paper is quite clear and concise, the English language and grammar could be improved in places.

Reviewer #2 (Remarks to the Author):

The authors reported an electricity-driven synthesis of triazolopyridinones from hydrazides. The latter is prepared readily from pyridyl hydrazines and carboxylic acids. The reactions are conducted with simple setups and do not require chemical oxidants or transition metal catalysts. The use of enantioenriched carboxylic acids results in enantioselective formation of heterocyclic products. The results of these useful reactions are unexpected and interesting. Overall, this work has been nicely carried out and well presented. This review recommends the publication of this work on Nat Commun.

1. For the reaction mechanism, direct oxidation of the substrates also occurs considering their relatively low oxidation potential and a moderate yield in the absence of the catalyst. The C-N bond formation and C-C migration most likely occur in a stepwise manner rather than in one concerted step, e.g. addition to the carbonyl followed by migration.
2. Have the authors tried diazine derived substrates to increase the number of nitrogen atoms on the heteroaromatic scaffold?
3. Continuous flow synthesis employs flow reactors. The use of a pump to recycle the reactant with a batch reactor (Fig 3d) should not be referred to as continuous flow synthesis.

Revised manuscript submission:

Manuscript ID NCOMMS-20-01691-T

General comments:

We are pleased that the Reviewers noted the novelty and potential utility of our described “Enantioselective Electrocatalytic Rearrangement for the Synthesis of Hindered Triazolopyridinone Derivatives”. The authors would like to express their gratitude to the Editor and Reviewers for their valuable comments and suggestions. All of the suggested modifications have been incorporated into this revised manuscript (highlighted in red). As a result, we believe that the concerns of the Reviewers have been addressed and the manuscript has been significantly improved.

Response to Reviewers:

Reviewer #1 (Remarks to the Author):

The submitted manuscript reports an electrochemical synthesis of *N*-alkyl triazolopyridinones from hydrazides of 2-hydrazinopyridine using a triarylamine as an electrochemical mediator, with around 60 reported examples. The triazolopyridine motif is found in a number of synthetic bioactive molecules that are of interest in drug discovery. To date, the synthesis of *N*-alkyl triazolopyridine containing compounds has been limited mostly to sterically unencumbered nitrogen substituents, installed largely through *N*-alkylation using primary alkyl electrophiles. The electrosynthesis described in the current work is effective for a variety of primary, secondary and tertiary *N*-substituents. Significantly, enantiomerically enriched hydrazide precursors possessing stereogenic centers alpha to the carbonyl group undergo stereospecific migration of the chiral group with retention of stereochemistry.

The reaction mechanism is intriguing, and the authors propose that the hydrazide intermediate is oxidized to the diazo intermediate (shown as *trans* stereoisomer), which they propose, cyclizes to a pyridinium intermediate. This undergoes a 1,2-shift

(C to N) of the acyl (COR) R substituent. A reactions were carried out to exclude mechanistic pathways involving radical, or carbocation intermediates. In one case an acyl diazo intermediate was isolated as a reaction side product, and shown to undergo rearrangement to the corresponding triazolopyridinone. In the electrochemical step, the two-electron oxidation is proposed to be mediated by the triarylamine. The process was performed on scales up to ~10 g, which should be attractive to those wishing to apply the methodology.

Overall, this is an intriguing electrochemical transformation, which should be of broad interest to organic chemists, particularly synthetic chemists and medicinal chemists. There are very significant elements of novelty and originality in the work, and the results would not have been easy to predict (I was not able to find related transformations). The mechanistic discussion is somewhat limited in the main paper, and the proposed cyclization of a trans-diazo species deserves further comment (barrier to cis/trans isomerization, or addition-elimination of MeOH?).

Currently we are still working on this related project, and we have no direct evidence to confirm the proposed mechanism. However, we did isolate and characterize the diazo intermediate 83, and confirmed that it can be converted into cyclized product 82. Also, the energy barrier from trans to cis-diazo species is low (around 3.6 kcal mol⁻¹ for similar compounds). Therefore, we believe the diazo species is one of the key intermediates for this transformation. The mechanism discussion detail has been modified in Page 8 (mian text), a cis-diazo intermediate III has been added in figure S6 in Page S53 (SI).

There is no addition-elimination of MeOH for this reaction, the methoxy anion produced in the cathode worked as base for the deprotonation of hydrazide.

While I was not entirely convinced by the mechanistic proposal, I am not able to propose a more convincing one. The role of the mediator is not commented on in the main paper, and it is not clear if the mediator or the supporting electrolyte are recovered.

The mediator trisarylamine worked as redox catalyst, which has been added in Page 4

in the main text. The mediator or the supporting electrolyte can be recovered after the workup, the details were added on Page S7 in SI.

These are significant points, and as the manuscript stands, the claims of eco friendly and sustainable should certainly be removed.

We agree that this reaction is not 100% green. However, in consideration of the atom economy of the key cyclization step, the recyclability of the solvent, mediator and electrolyte, more importantly the electricity was used as oxidant without the requirement of external harmful oxidants, we think this transformation is relatively environmentally friendly and sustainable.

However, that does not detract from the interest and novelty of the work, which I am sure will influence others working in electrosynthesis.

Overall, the paper should be published in Nature Communications subject to addressing some comments noted below.

1 I did not see any comment from the authors on *N*-aryl triazolopyridinones. Are these accessible using the reported method?

We tried the reaction with benzoyl hydrazide as substrate, unfortunately we didn't obtain the *N*-aryl triazolopyridinones. It might be because the benzoyl hydrazide didn't form the stable radical intermediate.

2 Abstract, line 18: "This enables the atom- and step-economical preparation of more than 60 functionalized triazolopyridinone derivatives under mild conditions in a sustainable manner." This statement is not really correct. The preparation of the starting materials is not atom-economical, so the whole sequence can not be claimed to be atom-economical, even if the electrosynthesis is. The particular example of electrosynthesis is actually not that great in terms of sustainability either, due to the large excess of added electrolyte included in the electrolysis. Furthermore, is the mediator recovered? It is a high molecular weight compound, so the mass used is probably not insignificant, even at 10 mol %.

The sentence has been modified as follows: This enables the facile and efficient

preparation of more than 60 functionalized triazolopyridinone derivatives under mild conditions in a sustainable manner.

The electrolyte could be recovered by recrystallization from the aqueous phase. The triarylamine and product which are in organic phase after workup can be separated and recovered easily by silica flash column chromatography. The details were added on Page S7 in SI.

3 Line 44-46, page 2. Improve accuracy of statement: “However, the atom-economy of these cross-coupling reactions is low because of the requirement of activating groups which were discarded as waste after the reaction.” The previous sentence describes Kolbe and Barton procedures for radical generation. The statement is certainly true for the Barton method, but less accurate for Kolbe (although it does lose CO₂). The next sentence, referring to Baron’s work on “electro generated carbocations from carboxylic acids” (ref 40), is then described as sustainable (this is also decarboxylative, but also contains additives such as silver salts). This statement described for activating groups such as Barton and NHPI ester procedure, not including Kolbe-type procedure which do not need activating group.

Thanks for pointing out this inappropriate description. It was corrected in the main text.

4 In Fig 1. B. Can the authors clarify whether they are referring to “2 chiral” or enantiomerically enriched examples? Please consider the use of the term “chiral”.

It was changed to “2 enantiomerically enriched examples”.

5 Figure 1: Is the process electrocatalysis? It is an electrochemical reaction mediated by an electrocatalyst.

It was changed to “electrochemical reaction”.

6 Figure 1: Is the process eco-friendly? The electrochemical process uses 5:3 ratio of supporting electrolyte to substrate, which is quite high. Was the Bu₄NBF₄

recovered after the reaction? The electrochemical reaction is also quite dilute (0.03 M), which requires quite a lot of solvent when scaled up. The triarylamine mediator is used in 10 mol %, but is that recovered?

After the reaction the solvent was recovered by reduced pressure, the crude was then extracted with ethyl acetate and washed with water. The $n\text{Bu}_4\text{NBF}_4$ can be recovered by recrystallization from water, and the triarylamine mediator and product could be separated and isolated easily by silica flash column chromatography. This statement was added as notes in page S7 in the supporting information.

7 Line 88, page 4: “This protocol also can be conducted in a simple one-pot reaction without isolation of the hydrazide though the overall yields (in the brackets) are a little bit lower.” The procedure described is not “one pot”; the authors simply took the crude material (containing the urea byproduct from DCC) from the first coupling reaction into the electrochemical step (solvent change and reaction vessel change).

It was changed as follows: This protocol also can be conducted without isolation of the hydrazide though the overall yields (in the brackets) are a little bit lower.

8 Line 119, page 6: “A library of previously inaccessible trazodone analogues could be made efficiently by this strategy, and further evaluated for the binding affinity for some subtypes of serotonin receptors in search of novel drugs for the treatment of depression and anxiety.” Suggest that this is revised to indicate that the methodology could be used for the synthesis of previously inaccessible trazodone analogues.

It was changed as suggested.

Although the paper is quite clear and concise, the English language and grammar could be improved in places.

The English language and grammar were improved further.

Reviewer #2 (Remarks to the Author):

The authors reported an electricity-driven synthesis of triazolopyridinones from hydrazides. The latter is prepared readily from pyridyl hydrazines and carboxylic acids. The reactions are conducted with simple setups and do not require chemical oxidants or transition metal catalysts. The use of enantioenriched carboxylic acids results in enantioselective formation of heterocyclic products. The results of these useful reactions are unexpected and interesting. Overall, this work has been nicely carried out and well presented. This review recommends the publication of this work on Nat Commun.

1 For the reaction mechanism, direct oxidation of the substrates also occurs considering their relatively low oxidation potential and a moderate yield in the absence of the catalyst. The C-N bond formation and C-C migration most likely occur in a stepwise manner rather than in one concerted step, e.g. addition to the carbonyl followed by migration.

To be honest we are not sure whether the reaction involves a stepwise or concerted manner. It might be proceeded via the nucleophilic addition of nitrogen to carbonyl group leading to a tetrahedral intermediate which then underwent an aza-alpha-ketol type rearrangement. We are still working on the similar project and try to understand the reaction mechanism during the future investigations. To avoid the dispute, we decided to delete the word "concerted".

2 Have the authors tried diazine derived substrates to increase the number of nitrogen atoms on the heteroaromatic scaffold?

We tried the reaction with 2-hydrazineylpyrimidine derived hydrazide as the substrate, unfortunately there was no reaction under standard conditions.

3 Continuous flow synthesis employs flow reactors. The use of a pump to recycle the reactant with a batch reactor (Fig 3d) should not be referred to as continuous flow synthesis

It was deleted.

REVIEWERS' COMMENTS:

Reviewer #1 (Remarks to the Author):

I am satisfied that the authors have adequately addressed, or responded to, the points raised by the reviewers, and can recommend that the manuscript is published in Nature Communications.

Revised manuscript submission:

Manuscript ID NCOMMS-20-01691A

General comments:

We are pleased that the Reviewers are satisfied with our previous revisions. Here, we would like to submit the final version of our manuscript by addressing the valuable suggestions from both the Editor and Reviewers. All the suggested modifications have been incorporated into this revised manuscript. As a result, we believe that the concerns have been addressed and the manuscript has been significantly improved.

Response to Reviewers:

REVIEWERS' COMMENTS:

Reviewer #1 (Remarks to the Author):

I am satisfied that the authors have adequately addressed, or responded to, the points raised by the reviewers, and can recommend that the manuscript is published in Nature Communications.

We appreciate the comments from the reviewers, and the further suggested changes have been corrected.